# Taming Data Chaos: Agentic Knowledge Warehousing for Contextual Intelligence

## Abstract

Information seeking can be viewed as bridging the knowledge gap between a query and its answer. While large language models (LLMs) perform strongly across diverse tasks, their capacity to fill this gap is bounded by pretraining data and deteriorates on queries requiring specialized or up-to-date knowledge. A common solution is to augment LLMs with external knowledge, either by injecting retrieved evidence into the context or by interleaving retrieval with reasoning. The former restricts exploration of layered dependencies, whereas the latter is constrained by context length, limiting both efficiency and scalability. Yet complex tasks often involve intricate dependencies and may require processing large volumes of raw text, under which both strategies become inadequate.

To tackle this bottleneck, we present Agentic Knowledge Warehouse (AWARE), a retrieval paradigm that transforms vast unstructured data into minimal, task-specific knowledge consumable by LLMs. Rather than simply returning raw information, AWARE curates knowledge through an agentic process that plans, explores, and synthesizes evidence into coherent context. Specifically, it organizes raw corpora with document-level gist memory for global coverage, applies diffusion-based exploration with vertical exploitation to recover layered dependencies, and employs map–reduce inspired synthesis to integrate large-scale evidence into a compact, LLM-ready context. This design enables both in-depth exploration and scalable integration, reconstructing the knowledge space needed to address task-specific knowledge gaps. Experiments on GAIA, WebWalker, and BrowseComp show that AWARE outperforms baselines, validating its effectiveness and generality. Our codes are available in this *anonymous repository*.

## 1 Introduction

Recently, large language models (LLMs) have excelled in information-seeking tasks, generating coherent responses to queries ranging from simple to complex reasoning (Ouyang et al., 2022; Gemini Team, 2025; DeepSeek-AI, 2025). Yet their knowledge, fixed to the training corpus, is limited in coverage and timeliness. As a result, performance might deteriorate on knowledge-intensive tasks requiring specialized or up-to-date information (Zhao et al., 2024b; Huang et al., 2025).

To mitigate this limitation, LLMs are often augmented with external sources such as the web or local knowledge bases, most commonly through retrieval-augmented generation (RAG), where retrieved information is injected into the model's context to ground responses beyond pretrained knowledge (Lewis et al., 2020; Gao et al., 2024). While effective in many cases, this pre-inference retrieval scheme often falls short on complex tasks that require multi-step reasoning to uncover interdependent evidence (Zhao et al., 2024a). Tool-integrated reasoning methods improve on this by interleaving retrieval with reasoning, allowing agents to iteratively refine queries, interact with tools, and synthesize evidence (Jin et al., 2025; Li et al., 2025c), but its reliance on in-context evidence makes it inefficient and limited in scalability as context length grows. Beyond these limitations, a more fundamental challenge remains: when LLMs rely on real-world data, they inevitably face *Data Chaos*, where the data sources involve long-form, heterogeneous, unstructured, noisy, and redundant content, such as web pages and PDF files (Zhu et al., 2024). In such settings, useful information is sparsely embedded within large volumes of retrieved text, leading to a very low signal-to-noise ratio. As a result, LLM context windows are quickly saturated, leaving it unable to isolate and assemble the crucial evidence needed to bridge the knowledge gaps. Effective information seeking

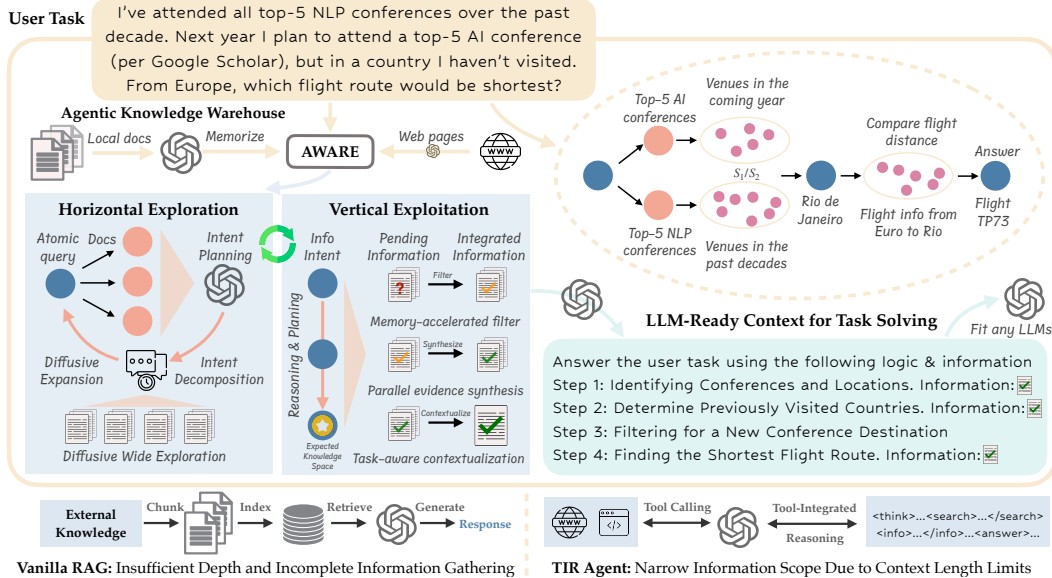

Figure 1: Illustration of a complex information-seeking task, where the answer depends on satisfying multiple conditions through horizontal exploration and vertical exploitation. AWARE addresses this agentically by formulating intents, decomposing them into atomic queries, and expanding coverage via diffusion to gather raw documents. Resolved intents advance iteratively to the next, with documents processed in parallel and synthesized through a map–reduce procedure into subspace knowledge, which is then transformed into an LLM-ready context.

therefore requires more than extracting isolated facts; it demands reorganizing fragmented raw text into coherent evidence that an LLM can reliably exploit (Zhang et al., 2024).

This challenge becomes particularly evident in complex tasks, which often require processing large volumes of raw text to uncover the information needed. As shown in Figure 1, identifying the "shortest flight route" requires reasoning under multiple constraints, with relevant evidence scattered across many web pages. Solving such tasks demands both vertical exploitation for depth and horizontal exploration for breadth. Classical RAG lacks the depth for multi-layered reasoning, whereas tool-integrated reasoning methods provide depth but is restricted in breadth by context length (Zhao et al., 2024a; Li et al., 2025c). In practice, addressing these tasks requires a new retrieval paradigm that can operate effectively under data chaos, distill task-relevant signals from large-scale raw text, and assemble a minimal yet sufficient evidence space to bridge the knowledge gap.

In this paper, we introduce _Agentic Knowledge Warehouse_ (AWARE), a retrieval framework that transforms vast unstructured sources into structured, task-specific knowledge consumable by LLMs. As shown in Figure 1, whereas single-pass retrieval and tool-integrated reasoning focus on how LLMs employ existing retrieval tools, AWARE instead reconceptualizes retrieval itself through an agentic process that organizes unstructured corpora into structured, task-specific knowledge and delivers it in an LLM-ready form. At the corpus level, AWARE ingests unstructured sources and constructs a structured warehouse. Each document is paired with a _gist memory_ that abstracts its overall theme and structure, and indexed through a hybrid scheme combining dense memory representations with sparse raw text for both global awareness and fine-grained access. At the task level, given a complex query, AWARE plans information intents, decomposes them into atomic sub-queries, and retrieves the corresponding atomic knowledge units. It expands coverage through diffusion-based horizontal exploration and resolves layered dependencies through vertical exploitation. To maintain efficiency at scale, this process leverages _memory-accelerated exploration_ to filter documents via gist memories and executes in a map–reduce–style parallel evidence synthesis outside the main reasoning loop. Finally, the collected evidence is synthesized into a coherent, task-specific knowledge chain and reformulated as a compact, LLM-ready context. Through this design, AWARE reconstructs the minimal sufficient knowledge space required to bridge complex information gaps, delivering it in a form that LLMs can readily comprehend and leverage at scale.

To evaluate the effectiveness of AWARE, we conduct extensive experiments on three challenging information-seeking benchmarks: General AI Assistant (GAIA), WebWalker, and BrowseComp. The results show that AWARE consistently outperforms baselines. Our contributions are threefold: (1) We introduce the Agentic Knowledge Warehouse, a new retrieval paradigm that transforms unstructured data into structured, task-specific knowledge in an LLM-ready form, seamlessly applicable to any standalone model. (2) We instantiate AWARE through a flexible framework that integrates document-level gist memory with an agentic decision process, combining diffusion-based horizontal exploration and vertical exploitation to realize contextual intelligence. (3) We provide extensive empirical validation showing that AWARE effectively connects large-scale external knowledge with LLM reasoning, offering a scalable and general solution to enhance standalone LLMs.

## 2 RELATED WORK

Incorporating large language models (LLMs) with external tools for knowledge augmentation has emerged as a crucial paradigm for extending their capabilities beyond static pretraining (Zhao et al., 2024b; Li et al., 2025d). Early studies explored retrieval-augmented generation (RAG) (Lewis et al., 2020), where an input query retrieves relevant evidence before inference, and the retrieved content is injected into the model's context (Zhao et al., 2024a). Subsequent enhancements have since been proposed, including query rewriting (Chan et al., 2024), self-critique mechanisms (Asai et al., 2024), memory augmentation (Qian et al., 2025), and graph-based retrieval strategies (Edge et al., 2024). While effective in many settings, these pre-inference schemes face limitations when information needs are multi-layered or sparsely embedded across sources (Zhang et al., 2024; Wei et al., 2025).

To address this, agentic search methods have recently gained traction. Most build on the ReAct framework (Yao et al., 2023), further optimized through expert-designed workflows (Li et al., 2025b; Chen et al., 2025; Qiu et al., 2025) or via end-to-end reinforcement learning (Jin et al., 2025; Sun et al., 2025; Shi et al., 2025). Beyond classical knowledge-intensive tasks such as those in Wikipedia-based datasets (Petroni et al., 2021), recent work has also shifted attention to challenging information-seeking benchmarks like GAIA and BrowseComp (Mialon et al., 2023; Wei et al., 2025), which demand deep reasoning and long-horizon planning. Representative approaches include TTD-DR (Han et al., 2025), WebThinker (Li et al., 2025c) and the WebAgent series (Li et al., 2025a; Geng et al., 2025; Wu et al., 2025a), which emphasize reasoning-intensive exploration across real-world web environments.

Overall, prior efforts have largely focused on how to leverage existing retrievers in different ways (Zhang et al., 2025). In contrast, our proposed AWARE establishes a new retrieval framework that directly constructs minimal yet sufficient knowledge for complex tasks, delivering curated LLM-ready context as a foundation for downstream reasoning.

## 3 METHOD

### 3.1 PRELIMINARY

**Complex Information-Seeking Task.** Solving a task with an LLM can be formalized as $\mathcal{Y} = \Theta(\mathcal{X} \mid \mathcal{K})$, where $\Theta$ denotes the model's generative function and $\mathcal{K}$ captures the knowledge required to bridge the gap between input and output. In this view, producing the correct answer amounts to filling a *knowledge gap* that separates $\mathcal{X}$ from $\mathcal{Y}$. When the task is simple fact-based or commonsense in nature, the gap is typically small and can often be resolved by the model's pretrained knowledge or a single retrieval step. In contrast, complex tasks create a much larger and more intricate gap. Recovering the expected knowledge space $\mathcal{K}$ in such cases is challenging, requiring multi-layered exploration and iterative decision-making in which evidence is progressively gathered, refined, and integrated until the gap is sufficiently closed to yield a reliable answer.

For *complex information-seeking tasks*, solving the problem typically unfolds as a multi-step reasoning process, where the required knowledge emerges in stages rather than all at once. In this setting, the knowledge gap $\mathcal{K}$ can be formally represented as a sequential *knowledge chain*:

$$\mathcal{K} = (\mathcal{K}_1 \rightarrow \mathcal{K}_2 \rightarrow \cdots \rightarrow \mathcal{K}_t), \tag{1}$$

where $\mathcal{K}_i$ denotes the crucial knowledge required at the $i$-th reasoning step, and the arrow $\rightarrow$ indicates the sequential dependency among steps. Each intermediate knowledge space $\mathcal{K}_i$ is itself

formed by combining multiple *atomic knowledge spaces*:

$$\mathcal{K}_i = \mathcal{S}_{i,1} \cap \mathcal{S}_{i,2} \cap \cdots \cap \mathcal{S}_{i,n_i}, \tag{2}$$

where each $\mathcal{S}_{i,j}$ represents a minimal unit of knowledge that can be directly retrieved through a single query $q$, typically consisting of a set of relevant documents $\{D\}^1$.

This formulation emphasizes two complementary dimensions of reasoning. *Depth* arises from the sequential composition of the knowledge chain, while *breadth* comes from the conjunction of multiple atomic knowledge spaces within each step. Special cases follow naturally: when $t = 1, n_i = 1$, the task reduces to a single-hop factual query; when $t > 1$ but each $n_i = 1$, it corresponds to simple multi-hop reasoning over independent facts.

From an information-theoretic perspective, the *knowledge gap* $\mathcal{K}$ quantifies the additional information required to determine the correct answer $\mathcal{Y}$ given input $\mathcal{X}$. Solving a complex task can thus be seen as an iterative reduction of conditional entropy,

$$H(\mathcal{Y} \mid \mathcal{X}) > H(\mathcal{Y} \mid \mathcal{X}, \mathcal{K}_1) > \cdots > H(\mathcal{Y} \mid \mathcal{X}, \mathcal{K}_1, \ldots, \mathcal{K}_t) = 0. \tag{3}$$

Here, *breadth* aggregates multiple atomic sources that jointly constrain uncertainty, while *depth* reflects the sequential dependencies through which uncertainty is progressively eliminated. In this view, the *knowledge chain* functions as an information channel that transmits the missing bits required to close the gap between input and answer.

**Data Chaos: The Bottleneck for LLM Context.** We refer to *Data Chaos* as the state in which essential knowledge is entangled within vast amounts of unstructured, redundant, and noisy data that an LLM cannot directly exploit. Suppose the universal knowledge space is denoted by $\mathcal{S}$. In principle, one could hope to isolate a *minimal but sufficient* subset $\mathcal{K}^* \subset \mathcal{S}$ that contains exactly the information required for producing the answer $\mathcal{Y}$ and nothing more. Such a representation would be ideal, as it would minimize entropy and maximize the signal-to-noise ratio of the input context.

In practice, however, retrieval for complex tasks produces large volumes of raw text (e.g., hundreds of web pages or thousands of PDF documents) that remain far from this ideal. The data are dominated by formatting artifacts, boilerplate language, and irrelevant content, with useful knowledge sparsely embedded within. Feeding such raw collections into an LLM is both inefficient and ineffective: the context window is quickly saturated, and the high entropy of the input obscures the information truly relevant to the task. In this sense, retrieved data in their raw form are not *LLM-ready*, but rather exemplify the disorder of *Data Chaos*.

### 3.2 THE PROPOSED METHOD: AGENTIC KNOWLEDGE WAREHOUSE

To tackle the challenge of Data Chaos, we propose *Agentic Knowledge Warehouse* (AWARE), a retrieval framework that iteratively explores and exploits external knowledge, distilling high-entropy and noisy retrieval results into a minimal yet sufficient knowledge space, which is then transformed into task-specific, LLM-ready context for enabling contextual intelligence. AWARE operates in two stages: *data indexing* and *agentic knowledge discovery*, which we detail in the following sections.

#### 3.2.1 DATA INDEXING WITH GIST MEMORY

Indexing real-world text data is challenging: document lengths vary from short news snippets to full academic papers, and structures are highly diverse, as in web pages with templates, metadata, and mixed formats. Direct applying dense retrieval to such raw text struggles under these conditions: for very long documents, it lacks global awareness because the encoder can only process limited windows; for structurally complex documents, it fails to capture implicit layout or organizational cues that are not explicitly encoded in raw text.

To overcome these limitations, AWARE introduces an intermediate representation that makes implicit global and structural information explicit. For each document $D$, a lightweight long-context model produces a textual abstraction $\mathcal{D}$ that verbalizes the document's high-level topics and structural cues while omitting details. For example, a journal issue page may be represented by its

---

[1]Here, a "document" is used in a broad sense and may refer to a web page, a PDF file, or a full text piece.

title, scope, and the categories of included articles, without enumerating the individual titles. This abstraction parallels the way humans process long texts: after reading, we tend to retain only a gist-level memory that preserves the overall theme and structure while discarding fine-grained details. We therefore refer to $\mathcal{D}$ as a form of *gist memory*. Unlike standard dense embeddings, which are typically derived from limited text windows and thus capture primarily local semantics, gist memory encodes a document's global theme and structure, such as topical hierarchy and organizational flow. This richer abstraction allows retrieval methods not only to locate broadly relevant documents but also to filter and prioritize them more effectively, accessing cues that would otherwise be overlooked by dense representations alone.

Indexing then proceeds in a hybrid manner. Each gist representation $\mathcal{D}$ is encoded into a dense vector $\mathbf{z}_D \in \mathbb{R}^d$ to capture global semantics, while the raw document $D$ is indexed using a sparse scheme to retain fine-grained evidence. Given a query $q$, relevance is computed as:

$$\text{Rel}(q, D) = \alpha \cdot \text{sim}_{\text{dense}}(q, \mathbf{z}_D) + (1 - \alpha) \cdot \text{sim}_{\text{sparse}}(q, D), \tag{4}$$

where $\alpha \in [0, 1]$ balances semantic coherence from dense matching with detail sensitivity from sparse matching. This *gist-memory based hybrid index* ensures that retrieval remains both globally AWARE and locally precise, enabling effective navigation of heterogeneous knowledge sources.

### 3.2.2 AGENTIC KNOWLEDGE DISCOVERY

By the definition of Eq. (1), the expected knowledge space $\mathcal{K}$ for a complex task cannot be obtained in a single step but must be assembled hierarchically. Specifically, $\mathcal{K}$ is composed of a sequence of subspaces $\{\mathcal{K}_1, \mathcal{K}_2, \ldots, \mathcal{K}_t\}$, where each $\mathcal{K}_i$ captures the evidence needed to resolve one stage of reasoning. In turn, every $\mathcal{K}_i$ is constructed from a collection of atomic knowledge spaces $\{\mathcal{S}_{i,1}, \mathcal{S}_{i,2}, \ldots, \mathcal{S}_{i,n_i}\}$, each of which can be retrieved by issuing a single query $q$.

AWARE constructs this knowledge in an agentic manner. Given a task $\mathcal{X}$, the system reasons over it to issue an initial information intent $I_1$, identifies the underlying knowledge gaps, and decomposes $I_1$ into atomic sub-queries:

$$I_1 \mapsto \{q_{1,1}, q_{1,2}, \ldots, q_{1,n_1}\}. \tag{5}$$

Each sub-query $q_{1,j}$ corresponds to a concrete retrieval action defined in Eq. (4), yielding an atomic knowledge space $\mathcal{S}_{1,j}$. Collectively, these atomic spaces form the subspace $\mathcal{K}_1$. Once $\mathcal{K}_1$ provides sufficient evidence to resolve $I_1$, the process advances to the next intent $I_2$, producing $\mathcal{K}_2$ in the same manner. This sequential procedure continues until all subspaces are constructed, yielding an approximation of the expected knowledge space $\mathcal{K}$ required to solve the task. This process can be expressed recursively as:

$$\mathcal{Y} = \Theta_t\Big( \cdots \Theta_2\big(\Theta_1(\mathcal{X} \mid I_1, \mathcal{K}_1) \mid I_2, \mathcal{K}_2\big) \cdots \Big| I_t, \mathcal{K}_t\Big). \tag{6}$$

where $I_t$ denotes the information intent at step $t$, $\mathcal{K}_t$ is the corresponding subspace constructed from atomic spaces, and $\Theta_t$ is the reasoning operation that advances once $I_t$ is resolved.

Building on this general definition, we instantiate AWARE with three core mechanisms: *Diffusive Wide Exploration* for knowledge coverage, *Memory-Guided Parallel Synthesis* for processing efficiency, and *Task-Aware Contextualization* for synthesizing LLM-ready context. These components will be introduced in detail below.

**Diffusive Wide Exploration.** A central challenge in constructing a subspace $\mathcal{K}_i$ for a given intent $I_i$ lies in *intent alignment*: the description of $I_i$ may be biased or incomplete, so its initial sub-queries may fail to cover the expected subspace. To mitigate this, AWARE employs a *Diffusion Search* strategy designed to maximize the coverage of intent-relevant knowledge. After executing the initial queries and obtaining atomic spaces, the agent evaluates whether the accumulated evidence suffices for $I_i$. If not, it expands the search frontier by generating additional queries conditioned on past results, thereby progressively enlarging the retrieved knowledge space:

$$\{q_{i,1}, \ldots, q_{i,n_i}\} \mapsto \{\mathcal{S}_{i,1}, \ldots, \mathcal{S}_{i,n_i}\} \mapsto \begin{cases} \mathcal{K}_i, & \text{if sufficient,} \\ \{q_{i,n_i+1}, \ldots\}, & \text{otherwise.} \end{cases} \tag{7}$$

This recursive expansion allows AWARE to iteratively refine and broaden the evidence pool, ensuring the resulting subspace $\mathcal{K}_i$ captures the full scope of knowledge necessary to resolve the intent.

**Memory-Guided Parallel Synthesis.** A second challenge is *scalability*: as diffusion expands, the number of queries and retrieved documents grows rapidly, making exhaustive analysis prohibitively costly. To address this, AWARE employs *memory-guided parallel evidence synthesis*, inspired by the map–reduce paradigm. Each retrieved document $D$, equipped with its gist memory $\mathcal{D}$, is first processed by a filtering operator $\mathcal{F}$ that performs lightweight relevance checks based solely on $\mathcal{D}$, discarding irrelevant candidates without accessing the full text. The surviving documents are then mapped in parallel by an extraction operator $\mathcal{E}$ into fine-grained evidence units, which are subsequently reduced by a synthesis operator $\mathcal{R}$ into the constructed subspace $\mathcal{K}_i$ for intent $I_i$:

$$\mathcal{K}_i = \mathcal{R}\Big(\{\mathcal{E}(D) \mid D \in \mathcal{F}(\{D\}, I_i, \{\mathcal{D}\})\}\Big). \tag{8}$$

All operators $\mathcal{F}, \mathcal{E}, \mathcal{R}$ are powered by an auxiliary lightweight LLM, enabling AWARE to filter aggressively, extract in parallel, and synthesize compactly. This design ensures scalability and efficiency while preserving broad evidence coverage with manageable reasoning cost.

**Task-Aware Contextualization.** After evidence collection, AWARE organizes the gathered information into a structured, task-specific context. Formally, given a task $\mathcal{X}$, the generated intents $\{I_i\}$ and the constructed subspaces $\{\mathcal{K}_i\}$, the system assembles an organized knowledge chain:

$$\mathcal{C} = \mathcal{X} \cup (I_1 \to \mathcal{K}_1) \to (I_2 \to \mathcal{K}_2) \to \cdots \to (I_t \to \mathcal{K}_t), \tag{9}$$

which explicitly encodes the reasoning trajectory and its supporting evidence. In essence, $\mathcal{C}$ represents the *LLM-ready form* of the expected knowledge space $\mathcal{K}$: compact, structured, and directly consumable by an LLM. This task-specific context can then be fed into any downstream LLM to generate the final answer $\mathcal{Y}$. We refer to this transformation from raw retrieval to structured, reasoning-ready context as AWARE's *contextual intelligence*.

In summary, AWARE is a retrieval framework that leverages an agentic paradigm to construct minimal yet sufficient LLM-ready knowledge representations for diverse tasks. Further implementation details are provided in Appendix A.1, and Table 2 illustrates the major processes via a case study.

## 4 EXPERIMENTS

### 4.1 DATASETS AND BASELINES.

**Datasets.** We evaluate AWARE on three challenging benchmarks for complex information-seeking. **GAIA** (General AI Assistant) comprises over 450 real-world queries spanning multi-step reasoning, multimodal understanding, and tool use (Mialon et al., 2023). Following prior work (Li et al., 2025c; Wu et al., 2025a), we use 103 text-only validation questions. **WebWalkerQA** includes 680 queries across domains such as conferences and organizations, requiring agents to traverse subpages and integrate dispersed evidence, which makes it a long-horizon reasoning challenge (Wu et al., 2025b). **BrowseComp** consists of 1,266 questions whose answers, although short and verifiable, are deliberately hidden beyond top search results (Wei et al., 2025). Because this benchmark is extremely difficult and often involves hundreds of page visits per query, it imposes substantial evaluation overhead. We therefore evaluate on two topics, *Art* and *History*, totaling 252 questions.

**Baselines.** We compare AWARE against three groups of baselines. (1) *Direct Reasoning*: strong standalone LLMs used without external tools, including Qwen2.5-32B, Qwen2.5-32B, QwQ-32B, GPT-4o, Gemini-2.5-Flash and DeepSeek-R1-671B (DeepSeek-AI, 2025; Gemini Team, 2025; OpenAI, 2024). (2) *Retrieval-Augmented Generation*: methods that inject retrieved evidence, such as vanilla RAG and enhanced variants with query planning or iterative refinement (Shao et al., 2023; Chan et al., 2024). (3) *Tool-Integrated Reasoning*: approaches that interleave retrieval with reasoning, including ReAct, Search-o1, and WebThinker (Yao et al., 2023; Li et al., 2025b;c). Appendix A.1 provides implementation details for AWARE and baselines.

### 4.2 MAIN RESULTS

Table 1 reports the performance of AWARE and baseline. Our key findings are as follows:

(1) Under direct reasoning without retrieval, all models handle GAIA tasks more readily, yet their accuracy remains modest. By contrast, accuracy drops sharply on WebWalkerQA and BrowseComp,

Table 1: Main experimental results. Best scores are shown in bold, and second-best are underlined. Following the official settings, we report Exact Match (EM) for GAIA and BrowseComp, and LLM Equivalence Accuracy for WebWalkerQA.

| Method | General AI Assistant | | | | WebWalkerQA | | | | BrowseComp | | |
|---|---|---|---|---|---|---|---|---|---|---|---|
| | Level 1 | Level 2 | Level 3 | Avg. | Easy | Medium | Hard | Avg. | Art | History | Avg. |
| *Direct Reasoning (w/o Retrieval)* | | | | | | | | | | | |
| Qwen2.5-32B | 20.5 | 9.6 | 8.3 | 13.6 | 3.8 | 2.5 | 3.3 | 3.1 | 0.0 | 0.0 | 0.0 |
| Qwen3-32B | 15.4 | 7.7 | 0.0 | 9.7 | 3.1 | 1.4 | 2.5 | 2.2 | 0.0 | 0.0 | 0.0 |
| QwQ-32B | 25.6 | 9.6 | 16.7 | 16.5 | 7.5 | 2.1 | 3.8 | 4.0 | 0.0 | 0.8 | 0.4 |
| GPT-4o | 23.1 | 15.4 | 8.3 | 17.5 | 6.7 | 6.0 | 4.2 | 5.5 | 0.8 | 0.8 | 0.8 |
| Gemini-2.5-Flash | 33.3 | 11.5 | 0.0 | 18.5 | 8.5 | 16.3 | 7.9 | 5.8 | 9.1 | 0.0 | 0.0 |
| DeepSeek-R1-671B | 43.6 | 26.9 | 8.3 | 31.1 | 5.0 | 11.8 | 11.3 | 10.0 | 0.0 | 0.0 | 0.0 |
| *Retrieval-Augmented Generation (RAG)* | | | | | | | | | | | |
| Vanilla RAG (Qwen2.5-32B) | 12.8 | 11.8 | 8.3 | 11.8 | 23.1 | 14.3 | 11.3 | 15.3 | 0.0 | 0.0 | 0.0 |
| Vanilla RAG (QwQ-32B) | 33.3 | 36.5 | 8.3 | 32.0 | 36.9 | 26.1 | 33.5 | 31.2 | 2.4 | 1.6 | 2.0 |
| Query Planning (Qwen2.5-32B) | 30.8 | 17.3 | 0.0 | 20.4 | 29.4 | 36.4 | 25.0 | 30.7 | 0.0 | 0.0 | 0.0 |
| Query Planning (QwQ-32B) | 48.7 | 25.0 | 8.3 | 32.0 | 28.8 | 35.7 | 30.8 | 32.5 | 1.6 | 0.8 | 1.2 |
| Iterative RAG (Qwen2.5-32B) | 35.9 | 19.2 | 8.3 | 24.3 | 30.6 | 35.7 | 25.4 | 30.9 | 0.0 | 0.0 | 0.0 |
| Iterative RAG (QwQ-32B) | 51.3 | 28.8 | 8.3 | 35.0 | 29.4 | 32.9 | 31.3 | 31.5 | 0.8 | 0.0 | 0.4 |
| *Tool-Integrated Reasoning (TIR)* | | | | | | | | | | | |
| ReAct (Qwen2.5-32B) | 46.1 | 44.2 | 8.3 | 40.7 | 44.3 | 46.7 | 29.2 | 38.4 | 0.0 | 0.0 | 0.0 |
| ReAct (QwQ-32B) | 48.7 | 34.6 | 16.7 | 37.8 | 35.6 | 29.1 | 13.2 | 24.1 | 0.8 | 0.8 | 0.8 |
| ReAct (GPT-4o) | 51.2 | 34.6 | 8.3 | 34.6 | 34.6 | 42.0 | 23.9 | 33.8 | 2.4 | 1.6 | 1.9 |
| Search-o1-32B | 53.8 | 44.2 | 16.7 | 39.8 | 43.1 | 35.0 | 27.1 | 34.1 | 1.6 | 2.4 | 1.9 |
| WebThinker-32B-Base | 53.8 | 44.2 | 16.7 | 44.7 | 47.5 | 41.1 | 39.2 | 41.9 | 2.4 | 2.4 | 2.3 |
| WebThinker-32B-RL | 56.4 | **50.0** | 16.7 | 48.5 | 58.8 | 44.6 | 40.4 | 46.5 | 2.4 | 3.1 | 2.7 |
| **AWARE** (QwQ-32B) | **61.5** | 46.2 | **33.3** | **50.5** | 53.1 | **55.0** | **50.8** | **53.1** | **8.7** | **8.0** | **8.3** |

confirming that these benchmarks demand recent and long-tail knowledge rarely captured in model parameters. Interestingly, Qwen3-32B, although more recent, underperforms both Qwen2.5-32B and QwQ-32B, suggesting that Qwen3's hybrid reasoning design might compromise efficacy. Based on these observations, we use Qwen2.5-32B and QwQ-32B as the backbone models for RAG and tool-integrated reasoning baselines.

(2) AWARE consistently outperforms not only vanilla RAG but also advanced variants that incorporate query rewriting or iterative refinement, validating the robustness of its retrieval paradigm. Unlike these pre-inference schemes that often leave evidence fragmented or incomplete, AWARE employs diffusion-based exploration and memory-guided synthesis to recover layered dependencies while filtering noise at scale. This design yields substantial gains on tasks that demand multi-hop reasoning and long-horizon synthesis, where RAG methods struggle to provide coherent context.

(3) AWARE surpasses TIR baselines, including workflow-based methods (e.g., Search-o1) and end-to-end optimized systems (e.g., WebThinker). Although WebThinker benefits from large-scale in-domain training, AWARE without task-specific optimization still outperforms WebThinker-Base across all dimensions and achieves dataset-level gains over WebThinker-32B-RL, falling only on two levels within the dataset hierarchy. This highlights AWARE's diffusive and parallel exploration, which enables broader coverage and reduces the risk of missing critical evidence.

## 4.3 ABLATION STUDY

AWARE is designed as an integrated retrieval framework, operating as a unified system with interdependent components, making it more meaningful to analyze as a whole rather than in isolation. Accordingly, our ablation study examines three dimensions: (1) the role of different LLMs as AWARE's central reasoning agent, (2) the generalizability of AWARE-generated context across diverse models, and (3) the dynamics of agentic retrieval, with a focus on diffusion search depth and evidence synthesis efficiency. Figure 2 summarizes the results, which we discuss below.

**Impact of Reasoning LLM Selection.** AWARE relies critically on the capabilities of its central reasoning model. As shown in Figure 2 (a), it consistently outperforms the tool-integrated reasoning baseline (Search-o1) across different LLMs, demonstrating the robustness of its design. Nonetheless, the strength of the reasoning model plays a decisive role. Reasoning-oriented models such as QwQ-32B and Qwen3-30B-A3B achieve the best results, clearly surpassing Qwen3-32B, a hybrid model with diluted reasoning capacity. When paired with Gemini2.5-Flash, AWARE also delivers

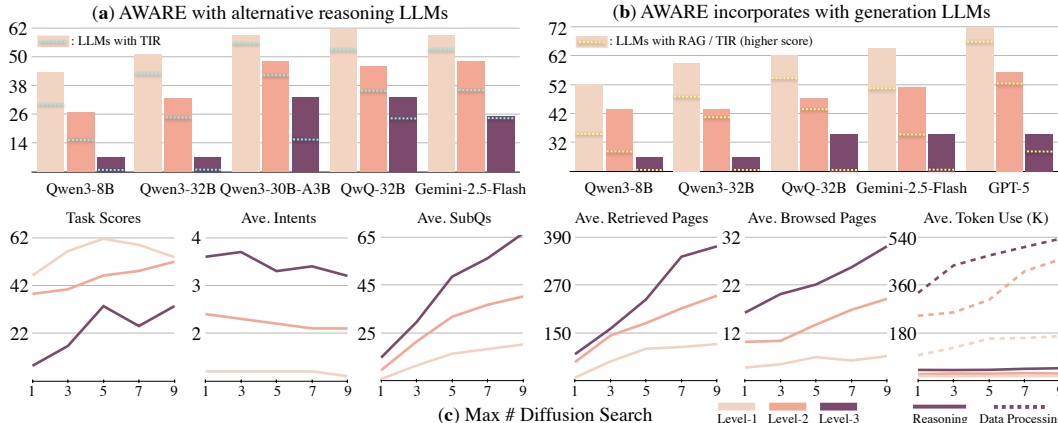

Figure 2: Analysis of AWARE on three perspectives: (a) effect of the central reasoning LLM, comparing AWARE with a TIR baseline (Search-o1); (b) transferability of AWARE's LLM-ready context, compared with RAG and TIR across downstream LLMs for answer generation; and (c) impact of the diffusion-search budget on performance and resulting retrieval dynamics.

competitive results through a dynamic strategy: the model enables "thinking mode" for complex planning steps while producing direct outputs for simpler ones, striking a balance between efficiency and accuracy. Overall, these findings show that while AWARE adapts well to diverse LLMs, its performance scales with the depth and quality of reasoning in the central agent.

**Generalizability of AWARE-Generated Context.** AWARE serves as a retrieval framework that produces reliable, task-specific context in an LLM-ready form, which can be seamlessly applied to any model. As illustrated in Figure 2 (b), supplying this curated context to different generation models consistently outperforms both TIR and RAG baselines, underscoring the robustness and generalizability of AWARE. For smaller models such as Qwen3-8B, the performance gains are especially pronounced, showing that AWARE can effectively compensate for the limited reasoning and knowledge capacity of lightweight LLMs. Conversely, when applied to stronger models such as GPT-5, the curated context is leveraged even more effectively, yielding further improvements and demonstrating AWARE's scalability across model strengths.

**Agentic Behavior across Diffusion Search Depths.** Our analysis in Figure 2 (c) highlights how AWARE's tailored techniques jointly contribute to effective and scalable retrieval. First, diffusion search proves critical: increasing its depth expands the evidence pool, improving task performance from depth 1 to 5 before fluctuating as information saturates. We also observe that deeper diffusion reduces the number of required search intents, easing the reasoning workload and accelerating convergence to answers. Second, deeper diffusion inevitably increases sub-queries and retrieved pages, especially for complex information-seeking tasks. Here, AWARE's memory-guided parallel evidence synthesis is validated: it filters out nearly 90% of irrelevant pages using gist memory and processes the remainder in a map–reduce manner, demonstrating strong scalability. Finally, token usage analysis shows that reasoning accounts for a small fraction of total cost compared to large-scale data processing. This validates AWARE's design of assigning heavy reasoning to strong central models while outsourcing bulk data handling to lightweight auxiliary models, achieving an effective balance between efficiency and performance.

## 4.4 CASE STUDY

Table 2 presents a case study that illustrates how AWARE constructs an LLM-ready context for a complex task. The query requires first identifying a *scientific genus* and then consulting academic papers, with the final answer derived by intersecting the animals discussed across these sources.

AWARE begins by leveraging the foundation model's intrinsic knowledge to establish the target genus, which anchors subsequent information intents. For the first intent, retrieving the precise titles

Table 2: Case study on a Level-3 sample from GAIA. The reasoning agent within AWARE first addresses the initial knowledge gap using intrinsic knowledge, then conducts agentic external knowledge exploration. The resulting task-specific context reconstructs a minimal yet sufficient knowledge space and delivers it in an LLM-ready form for solving the input task.

---

**Task**: What animals that were mentioned in both Ilias Lagkouvardos's and Olga Tapia's papers on the alvei species of the genus named for Copenhagen outside the bibliographies were also present in the 2021 article cited on the alvei species' Wikipedia page about a multicenter, randomized, double-blind study?  **Ground-truth answer**: Mice

---

**Initial Reasoning**: Identify the scientific genus named for Copenhagen. $\rightarrow$ Hafnia

**Information Intent $\mathcal{I}_1$**:
- Find scientific papers by Ilias Lagkouvardos concerning Hafnia alvei.
- Find scientific papers by Olga Tapia concerning Hafnia alvei.
- Find the Wikipedia page for Hafnia alvei to locate the 2021 article cited about a multicenter, randomized, double-blind study.
*4 atomic queries issued $\rightarrow$ 36 pages retrieved $\rightarrow$ 13 pages browsed; Tokens: 8.9K (reasoning), 227.2K (processing).*

**Knowledge Subspace $\mathcal{K}_1$**: Ilias Lagkouvardos authored a paper "An Insight into Goat Cheese: The Tales of Artisanal and Industrial Gidotyri Microbiota" on H. alvei were identified, while Olga Tapia authored a 2022 paper in Molecules titled "The Molecular Weaponry Produced by the Bacterium Hafnia alvei in Foods", discussing the bacterium's role in food systems. The Wikipedia page for Hafnia alvei cites a 2021 multicenter, randomized, double-blind study published in Nutrients—"The probiotic strain H. alvei HA4597® improves weight loss in overweight subjects under moderate hypocaloric diet".

**Information Intent $\mathcal{I}_2$**:
- Search for the full text of Olga Tapia's 2022 paper "The Molecular Weaponry Produced by the Bacterium Hafnia alvei in Foods" published in Molecules and identify any animals mentioned within the text (excluding bibliographies). ...... [omitted]

---

**Task-Specific Context $\mathcal{C}$**: The task requires the following steps and information:

Step 1: Identify the scientific genus named for Copenhagen and confirm the relevant alvei species. Info: The genus is Hafnia, derived from the Latin name for Copenhagen; relevant species: Hafnia alvei.

Step 2: Identify the 2021 article cited on the Hafnia alvei Wikipedia page. Info: Nutrients (2021): "The probiotic strain H. alvei HA4597® improves weight loss in overweight subjects under moderate hypocaloric diet" (multicenter, randomized, double-blind).

...... [omitted]

Step 5: Identify animals mentioned in the 2021 Nutrients study. Info: Human participants (n=236); also referenced preclinical studies using obese mice (ob/ob, HFD-fed).

Step 6: Determine overlap of animals across sources. Info: The only animal shared in the papers is mice.

---

of the three papers, AWARE executes diffusive parallel exploration: issuing multiple atomic queries, gathering 36 candidate pages, filtering them with gist-level relevance checks, and browsing only 13 to distill the relevant evidence. This step demonstrates AWARE's ability to maximize coverage while keeping processing efficient. The process then advances to the next intent, shifting from paper discovery to full-text analysis, with new sub-queries generated adaptively to uncover the animals mentioned. Once all necessary evidence is accumulated, AWARE synthesizes the results into a structured, task-specific context that integrates both retrieved knowledge and intermediate reasoning steps. The resulting representation is compact yet sufficient, capturing logical dependencies across intents, minimizing redundancy, and fitting neatly within the LLM's context window.

Notably, the case highlights AWARE's efficiency: reasoning consumes only 8.9K tokens, while large-scale evidence processing consumes 227K tokens. This demonstrates AWARE's balanced design, where strong central models focus on reasoning while lightweight auxiliaries handle bulk text processing. As a result, AWARE achieves efficient exploration of massive raw text, effective evidence consolidation, and robust downstream reasoning without overloading the model with noise.

## 5 CONCLUSION

In this work, we introduced the Agentic Knowledge Warehouse (AWARE), a retrieval paradigm that enriches LLMs with external knowledge in a structured, task-specific form. At the corpus level, AWARE abstracts vast unstructured sources into gist memories, offering global semantic coverage and encoding implicit structural cues often missed by conventional dense retrieval. At the task level, it engages in an agentic reasoning process that decomposes complex queries into layered intents, conducts diffusion-based horizontal exploration with vertical exploitation, and synthesizes the results into a coherent, compact, and LLM-ready context. This design enables AWARE to reconstruct the minimal yet sufficient knowledge space needed to close task-specific knowledge gaps.

Extensive experiments on challenging information-seeking benchmarks, complemented by ablation studies across multiple dimensions and a detailed case study, validate the soundness, scalability, and adaptability of AWARE. Taken together, these results show that AWARE constitutes a practical, robust, and broadly applicable solution for enabling contextual intelligence in standalone LLMs. A discussion of limitations and future directions is provided in Appendix A.2.

## REPRODUCIBILITY STATEMENT

Our method is implemented using open-source models (Qwen series), open frameworks (Lang-Graph), and publicly available datasets. All codes and prompts are released in an anonymous repository. The repository also contains experiment logs with intermediate outputs, including search intents, atomic queries, browsed web pages, refined evidence, and organized information. These records facilitate reproduction of our results and provide additional case studies that demonstrate the behavior of our approach. We also provide implementation details in Appendix A.1.

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

## A  APPENDIX

### A.1  IMPLEMENTATION DETAILS

In the main experiments, AWARE adopts QwQ-32B as the central reasoning model, supported by Qwen2.5-7B as an auxiliary processor for parallel data synthesis. For each query, AWARE curates a task-specific context, which is then directly fed into standalone LLMs for answer generation. Unless otherwise specified, the maximum diffusion depth for the *diffusive wide exploration* is set to 5.

To construct the web page collection used in the benchmarks, we first run AWARE directly with a search engine. In this initialization step, the local index is replaced by the search engine, and the use of gist memory is approximated by the first 1,024 tokens of each retrieved page. For every query, the top-20 web pages are collected. After five full runs on each benchmark, this procedure yields approximately 100K web pages in total. These pages are then processed to generate gist memories, which serve as the basis for the subsequent indexing process within AWARE.

Notably, evaluation with the online search engine proves both *slow* and *unstable*, since each sample requires executing multiple sub-queries and crawling tens to hundreds of web pages. The performance in this setting is also substantially lower than that achieved with the local index.

During the *data indexing process*, we employ BGE-M3 as the dense embedding model (Chen et al., 2024), complemented by a BM25 index constructed over the full web content. All retrieval operations are instantiated using ElasticSearch, which provides a stable and scalable infrastructure for large-scale search. For online retrieval, we rely on Google's Custom Search JSON API to identify relevant pages, and utilize Jina AI's Web Reader to extract full web content. For all baselines, we either report results directly from their original papers or reproduce them using official implementations. All experiments are conducted on a node of eight NVIDIA A100-40G GPUs.

To ensure transparency and reproducibility, we release all prompts used in AWARE along with full experiment logs, including intermediate artifacts such as search intents, atomic queries, retrieved

and browsed pages, refined evidence, and organized knowledge. These materials are available in *this anonymous repository*.

## A.2 LIMITATIONS AND FUTURE DIRECTIONS

Although AWARE demonstrates strong performance across diverse benchmarks, several limitations of this work should be acknowledged.

**Method Scope.** AWARE is proposed as a general retrieval paradigm that operates independently of specific models and can integrate seamlessly with both open-source and closed-source LLMs. However, unlike data-driven approaches that train task-specific models, AWARE does not incorporate optimization strategies tailored to particular domains. This limitation arises from objective constraints. The agentic framework of AWARE involves multiple capabilities such as planning, intent generation, evidence refinement, and synthesis, all of which would require carefully annotated or synthetically generated data for supervised optimization. While reinforcement learning could serve as an end-to-end alternative, producing large-scale, high-quality training data and running optimization for large models would demand significant resources. For example, reinforcement learning on a 32B model reasonably requires at least 32 H100 80G GPUs, which remain beyond reach.

Despite this, we argue that AWARE can continually benefit from improvements in general-purpose LLMs. The very skills required within AWARE, including reasoning, planning, and synthesis, fall within the optimization scope of mainstream models. Thus, even without explicit task-specific fine-tuning, AWARE achieves strong results. Moreover, excessive specialization on narrow domains may harm the generality of large models, introducing overfitting risks and reducing adaptability.

**Experimental Scope.** Given that AWARE is currently designed for text-only scenarios, we follow prior work and exclude the non-textual portion of GAIA, which prevents us from evaluating on the full benchmark. For BrowseComp, the evaluation is further constrained by the high cost of API usage. Even when restricted to two topics, *Art* and *History*, constructing the corresponding web page collection required crawling nearly 50,000 pages and issuing close to 10,000 Google queries. Across all three benchmarks in this paper, the combined use of the Google Search API and the Jina web page crawling API has already incurred costs exceeding 1,200 USD, placing considerable pressure on the experimental budget and limiting our ability to scale the evaluation further. Future work may explore more cost-efficient pipelines to enable broader coverage of these benchmarks.

**Baseline Coverage.** We strive to ensure that baselines in our main experiments are comparable in model size, open-sourcing status, and implementation feasibility. Nonetheless, it is not possible to evaluate against all related baselines. Some rely on substantially different model sizes, others are not fully released, and some require resources that are unavailable in our setting.

**Future Directions.** These limitations primarily reflect constraints in data availability, computational resources, and experimental scope, rather than methodological shortcomings. Addressing them naturally opens several avenues for future work. One direction is to explore lightweight optimization strategies, such as reinforcement learning with synthetic data, to adapt AWARE more closely to domain-specific tasks. Another is to extend AWARE beyond the text-only setting toward multimodal benchmarks, where information is distributed across heterogeneous modalities. Finally, exploring more precise strategies for constructing web page collections would reduce the crawling of irrelevant pages, thereby lowering overall API costs and enabling more efficient evaluation.

## B AI USAGE DISCLOSURE

In this work, AI assistants were used exclusively for polishing the manuscript, including grammar checking and language refinement. The initial draft was prepared manually by the authors, and only selected sections were refined with AI assistance.

AI assistants did not contribute to any other part of the research, including ideation, literature review, or figure preparation.

