# OpenReview forum: "Taming Data Chaos: Agentic Knowledge Warehousing for Contextual Intelligence"
_ICLR.cc/2026/Conference — ICLR 2026 Conference Withdrawn Submission_

### Official Review · Reviewer_yGAh · 2025-10-16

**Soundness:** 2
**Presentation:** 2
**Contribution:** 2
**Rating:** 4
**Confidence:** 4

**Summary:**

This paper proposes AWARE, which refines the indexing of corpora to be retrieved and leverages diffusion-search to augment the reasoning process of LLMs to solve complex problems.

**Strengths:**

- This paper proposes a training-free method to achieve better performance compared with other baselines.
- This paper ablates on its several core parts and test on several complex benchmarks to prove its generability.

**Weaknesses:**

- Some implementation details are missed, e.g., how do you decide the value of \alpha, and what value do you use in your experiment? Also, what is the reasoning and data processing in Figure 2c?
- The construction process seems vague to me. How do you obtain the web pages you need to answer the question? Could you provide a case study for construction? From where I stand, the LLM firstly generates question-relevant sub-queries and gathers corresponding web pages, and splits them chunk by chunk to further process them to be gist-memory. Is it correct?
- The conclusion of the first ablation is not correct. Indeed, Qwen3-32B achieves better performance on various reasoning benchmarks compared with QwQ-32B, and Qwen3-30B-A3B can also disable thinking mode.
- Since all the processing leverages Qwen2.5-7B as the core part, human validation of the process should be provided. And as it can be viewed as an agent system, it is important to break the system into pieces to analyze each part.

**Questions:**

- For benchmark choices, why do you sample two subjects of BrowseComp rather than randomly sample?

---

### Official Review · Reviewer_kFTX · 2025-10-29

**Soundness:** 2
**Presentation:** 3
**Contribution:** 2
**Rating:** 4
**Confidence:** 3

**Summary:**

The paper introduces a compact retrieval framework designed to overcome the limitations of traditional retrieval-augmented generation (RAG) and tool-integrated reasoning (TIR) methods in large language models (LLMs). AWARE addresses the issue called *Data Chaos*, where relevant knowledge is buried within raw, noisy, and unstructured data. AWARE organizes this data into a structured, task-specific knowledge format that is LLM-ready. The paper details how AWARE uses multiple agentic processes to integrate external knowledge. Experimental results on benchmarks like GAIA, WebWalkerQA, and BrowseComp show that AWARE significantly outperforms simple RAG and TIR methods.

**Strengths:**

1. **Integration of multiple agentic modules**: The framework combines several agentic processes (planning, exploration, synthesis) to tackle the problem of Data Chaos, which presents a complete, multi-faceted approach.
2. **Methodology clarity**: The paper explains technical aspects clearly, and the figures and case studies provide helpful visual explanations of the method and its workings.
3. **Cost & model analysis**: The paper provides a detailed analysis of the computational costs associated with AWARE and substitute different LLMs, giving readers a clear understanding.

**Weaknesses:**

1. **Multi-facility & related work**: While the paper introduces an complete framework, multiple jargon-heavy components have been mentioned. The components cover a wide range of techniques: unstructured indexing, query expansion, and LLM-friendly context synthesis for large-scale retrieval. Each of these components have its counterparts in prior work [1,2,3] which should be involved in the discussion and experiments.
2. **Resource intensity**: The use of multiple models and the reliance on external services for web crawling can lead to high computational costs and delays, which have been mentioned for AWARE. However, the cost analysis should also cover baseline methods for a fair comparison.
3. **Ablation study**: The paper lacks an ablation study to isolate the contributions of each component (planning, exploration, synthesis) to the overall performance.
4. **Baseline selection**: Comparing AWARE only with simple RAG and TIR methods may not fully demonstrate its advantages. More advanced and multi-faceted RAG frameworks with different "LLM-friendly context design" [4,5] should be included as baselines to provide a more comprehensive evaluation.


[1] Huang et al. KET-RAG: A Cost-Efficient Multi-Granular Indexing Framework for Graph-RAG. KDD 2025.
[2] Song et al. A Survey of Query Optimization in Large Language Models. 2024.
[3] Liu et al. Scaling External Knowledge Input Beyond Context Windows of LLMs via Multi-Agent Collaboration. 2025.
[4] Guo et al. LightRAG: Simple and Fast Retrieval-Augmented Generation. EMNLP 2025.
[5] Liang et al. KAG: Boosting LLMs in Professional Domains via Knowledge Augmented Generation. 2024.

**Questions:**

I found the overall motivation is a bit unclear. The paper initially argues about the bottleneck of raw RAG content, but later the experiments did not involve different transformations of the raw data. What should be the expected content formats for RAG content from different sources (e.g., web pages, documents, databases)? Should they be different or unified? And how could we measure the "LLM-readiness" of such content besides solely looking at the final performance on QA tasks?

---

### Official Review · Reviewer_9iYz · 2025-11-01

**Soundness:** 2
**Presentation:** 3
**Contribution:** 2
**Rating:** 4
**Confidence:** 4

**Summary:**

This paper addresses complex information-seeking tasks, where LLMs need to find and organize information through multi-step reasoning, cross-document integration, and external knowledge retrieval. The authors propose a retrieval framework called Agentic Knowledge Warehouse (AWARE). Specifically, AWARE transforms large-scale unstructured text into task-specific knowledge consumable by LLMs through a structured multi-stage pipeline. First, it generates "gist memory" summaries for each document to support subsequent hybrid retrieval. Then, based on the LLM's assessment of current information coverage, the system adaptively generates new retrieval queries using a "diffusive exploration" strategy to progressively expand the search scope and supplement relevant information. Next, it uses the model to filter, extract, and synthesize retrieval results. Finally, it organizes the relevant content into structured context for LLM reasoning and generation. The method is validated on three complex information retrieval benchmarks and demonstrates strong empirical performance.

**Strengths:**

- The proposed method presents a well-designed end-to-end system, providing a complete pipeline from data indexing to knowledge discovery, which appears reasonably reproducible. I appreciate the use of the Gist Memory concept, which introduces document-level summary representations that consider global semantics and structure, combined with mixed dense and sparse retrieval. These design choices address some practical computational cost concerns.
- The paper conducts comprehensive experiments across multiple benchmarks, comparing against various baselines and demonstrating strong empirical performance. The analysis section provides multi-dimensional evaluations to validate the method's effectiveness, including analyses of model types, context transferability, and search depth.

**Weaknesses:**

- While I find the system design reasonable, I believe the algorithmic innovation is quite limited. The work feels more like a combination of existing ideas into an LLM-based retrieval system rather than introducing novel methodological contributions. For example, using abstractive summarization for retrieval is not new and has been explored in prior work. Dense + sparse hybrid retrieval is already widely adopted in information retrieval, and the paper does not propose new fusion strategies or theoretical insights. Additionally, I find the term "Diffusion Search" somewhat misleading. It essentially describes iterative query expansion/refinement, and the "Diffusion" terminology may confuse readers (suggesting connections to diffusion models). Similar mechanisms have been extensively used in query reformulation and conversational search. The overall pipeline does not differ substantially from existing agentic open-domain retrieval frameworks; the differences mainly lie in engineering implementation choices.
- Many important implementation details are missing from the experiments. For instance: How is Gist Memory generated? Which model and prompts are used? How are intents formulated? What heuristics are employed? When does diffusion stop? What are the convergence criteria? These details are crucial for understanding and reproducing the work.
- The evaluation metrics appear limited, focusing primarily on end-task performance (e.g., exact match) without assessing the quality of the retrieval itself, such as relevance and coverage. The quality of the generated context (compactness, sufficiency) is also not evaluated. Furthermore, since the paper focuses on pipeline design, I believe it should report computational costs such as inference time and API call expenses to compare efficiency against baselines.
- The paper lacks a dedicated "Error Analysis" section. The authors do not systematically analyze error types, failure reasons, or typical failure cases, which would provide valuable insights into the method's limitations and potential improvements.

**Questions:**

AWARE uses an offline-constructed web page collection, while online baselines (such as Search-o1) require real-time search. Is this comparison fair?

---

### Official Review · Reviewer_F3me · 2025-11-01

**Soundness:** 2
**Presentation:** 3
**Contribution:** 2
**Rating:** 4
**Confidence:** 4

**Summary:**

The paper proposes AWARE, a framework designed to improve large language models on complex retrieval and reasoning tasks. The key idea is to first transform raw external data into a structured knowledge repository, then construct a minimal sufficient context for the model. The framework integrates gist-memory indexing, diffusion-style query exploration, and parallel evidence aggregation. Experiments on several benchmarks show improvements over standard RAG and tool-augmented reasoning baselines.

**Strengths:**

1. The paper clearly identifies the limitations of RAG and tool-integrated reasoning, and proposes a new framework to address them.
2. Experiments across different model scales and benchmarks demonstrate the effectiveness and robustness of the method.
3. The paper is well written, clearly structured, and easy to follow.

**Weaknesses:**

1. Unclear description of the Gist Memory. During corpus construction, the gist representation is approximated by the first 1k tokens, which is a standard dense embedding. The paper does not specify how it later “encodes a document’s global theme and structure.”
2. Limited novelty. While the framework is well organized, most of its main components are conceptually similar to existing ideas in prior work, such as Diffusive Wide Exploration, which is similar to performing iterative retrieval over subtasks.
3. Potential experimental bias. The corpus is built from five rounds of online retrieval, so most documents are already deemed relevant by the search engine. This setup may inflate the measured performance, as the authors note that direct online search performs worse. Moreover, comparisons with baselines like WebThinker, which use live web search, might not be entirely comparable since AWARE relies on a pre-filtered subset.

**Questions:**

1. What are the detailed steps for constructing the Gist Memory?
2. For the BrowseComp dataset, using a randomly sampled subset might be more representative and comparable than selecting specific categories.
3. In the experiments, do all datasets share the same set of webpages, or is a separate corpus built for each dataset?

---

### Note · Authors · 2025-12-19

I have read and agree with the venue's withdrawal policy on behalf of myself and my co-authors.